# Prenatal Diagnosis of Talipes Equinovarus by Ultrasound and Chromosomal Microarray Analysis: A Chinese Single-Center Retrospective Study

**DOI:** 10.3390/genes13091573

**Published:** 2022-09-01

**Authors:** Ruibin Huang, Xin Yang, Hang Zhou, Fang Fu, Ken Cheng, You Wang, Chunling Ma, Ru Li, Xiangyi Jing, Jin Han, Li Zhen, Min Pan, Dongzhi Li, Can Liao

**Affiliations:** 1Prenatal Diagnostic Center, Guangzhou Women and Children’s Medical Center, Guangzhou Medical University, Guangzhou 510620, China; 2School of Medicine, South China University of Technology, Guangzhou 510641, China; 3The First Clinical Medical College, Southern Medical University, Guangzhou 510515, China

**Keywords:** talipes equinovarus, prenatal diagnosis, chromosomal microarray analysis, fetal medicine

## Abstract

Background: There are few studies on the detection rate by chromosomal microarray analysis (CMA) of the prenatal diagnosis of talipes equinovarus (TE) compared to conventional karyotyping. We aimed to explore the molecular etiology of fetal TE and examine the detection rate by CMA, which provides more information for the clinical screening and genetic counseling of TE. Methods: In this retrospective study, pregnancies diagnosed with fetal TE were enrolled and clinical data for all cases were retrieved from our medical record database, including demographic data for pregnancies, ultrasound findings, karyotype/CMA results, and pregnant and perinatal outcomes. Results: Among the 164 patients, 17 (10.4%) clinically significant variants were detected by CMA. In 148 singleton pregnancies, the diagnostic rate of clinically significant variants was significantly higher in the non-isolated TE group than in the isolated TE group (10/37, 27.0% vs. 6/111, 5.4%, P < 0.001). In twin pregnancies, 1 (6.3%) pathogenic copy number variant was present in the other 16 twin pregnancies. Conclusions: This study demonstrates that CMA is useful for the prenatal genetic diagnosis of fetal TE. Fetal TE with the associated structural malformation correlates with a higher probability of clinically significant variants. This data may aid prenatal diagnosis and genetic counseling for fetal TE.

## 1. Introduction

Talipes equinovarus (TE) is the most common congenital malformation of the foot [1]. Its incidence is about 1–3 in 1000 of live births, and the proportion of male to female fetuses affected is about 2:1 [2]. TE includes four elements: metatarsus adductus, cavus foot, heel varus, and equinus, which is detected antenatally in over half of these cases by fetal ultrasound [3]. It can be unilateral (30–40%) or bilateral (60–70%) and can be either an isolated deformity (50–70%) or a manifestation of chromosomal abnormalities and other genetic syndromes (30–50%) [4]. Its etiology is not yet clear, but genetic and environmental factors are known to play an important role. Despite a high prevalence of TE, only a few pathogenic genes are known. The *PITX1*, *IGFBP3*, *TBX4*, and *RBM10* genes have been found to be associated with TE [5,6,7]. Although some TE fetuses, known as positional TE, return to a normal position, this is associated with intrauterine factors that limit fetal movements, such as oligohydramnios, twins, and uterine malformations [8]. Therefore, assessing the fetal genetic and clinical prognosis in fetuses with TE is essential.

In the past two decades, chromosomal microarray analysis (CMA) has been well studied and utilized in exploring genomic changes in fetuses with structural anomalies sonographically identified in prenatal settings. About 6% of fetuses with abnormal ultrasonography and normal karyotype can be detected with clinically significant chromosomal variations through CMA testing [9]. Amihood et al. [10] performed CMA testing in 269 prenatal cases of singleton pregnancies with TE in 2020 and detected 16 (5.9%) clinically significant variants. In contrast, Alvarado et al. [11] used CMA for genetic testing in 413 postnatal cases with isolated TE in 2013, and clinically relevant variants were identified in 2.4% of cases. Compared to postnatal studies, the detection rate of CMA in patients with prenatal ultrasound findings of TE is higher, and as the phenotype of TE may be a diagnostic clue for certain fetal syndromes, we believe that CMA may be necessary. However, there are only a few studies of the detection rate of CMA in the prenatal diagnosis of TE.

In this study, we review the clinical and molecular findings of 164 Chinese patients diagnosed with fetal TE at our center to explore its molecular etiology and examine the detection rate of TE by CMA, which may provide more information for clinical screening and genetic counseling.

## 2. Materials and Methods

### 2.1. Study Cohort

This was a retrospective cohort study by reviewing all prenatal cases of fetal talipes equinovarus diagnosed at the Prenatal Diagnosis Center, Guangzhou Women and Children’s Medical Center, from July 2013 to January 2022. All cases underwent a routine ultrasound scan for fetal anatomy, and associated abnormalities were recorded. Throughout the examination, if two long bones of the lower leg (tibia and fibula) were seen in the same plane as the sole, the diagnosis of fetal talipes equinovarus was made [9]. We divided pregnancies into unilateral or bilateral groups and isolated or non-isolated groups according to the type of TE. All parents of fetal TE received genetic counseling, which included the potential risks of invasive surgery and the possible implications of the findings, by the Maternal–Fetal Medicine team at our center.

We reviewed clinical data from all cases in our medical record database, including demographic data for pregnancies, indications for invasive examinations, ultrasound findings, karyotype/CMA results, and outcomes of pregnancy. Pregnancy outcomes are recorded partly autopsy results after the termination of pregnancy and partly by telephone or case review, focusing on clinical outcomes, gestational age at birth or termination of pregnancy, neonatal sex, presence of talipes equinovarus, other abnormalities, et al. This study was approved by the Ethics Committee of Guangzhou Women and Children’s Medical Center. Informed consent was obtained from the pregnant women before the invasive procedure.

### 2.2. Chromosomal Microarray Analysis

At our prenatal diagnostic center, CMA has replaced karyotyping as a first-line method for detecting fetal structural abnormalities since 2013. Genomic DNA was extracted from chorionic villi, amniocytes, cord blood using the Qiagen DNA Blood Midi/Mini kit (Qiagen GmbH, Hilden, Germany) according to the manufacturer’s protocol. Informed consent was taken to obtain a parental blood sample in order to run a trio analysis. We analyzed submicroscopic genomic imbalances using whole-genome high-resolution microarray analysis with CytoScan HD arrays and CytoScan 750 K arrays (Affymetrix, Santa Clara, CA, USA) according to the manufacturer’s protocols. The built reference genome was aligned on GRCh37/hg19. CytoScan 750K or CytoScan HD arrays are used to detect whole genome copy number variants (CNVs), as well as loss of heterozygosity (LOH) and isodisomy of uniparental disomy (iso-UPD), and to detect mosaicism at >30%. The mean turnaround time (TAT) for CMA from uncultured specimens was seven days. The process has been described in detail elsewhere [12].

Data were analyzed following American College of Medical Genetics guidelines, which categorizes all selected variants as pathogenic (P), likely pathogenic (LP), variants of unknown significance (VOUS), likely benign, or benign [13].

### 2.3. Statistical Analysis

Statistical analyses were performed using SPSS 25.0 (IBM, Armonk, NY, USA). Chi-square test or Fisher’s exact test was used to compare the characteristics among these subgroups. A *p* value of < 0.05 was considered statistically significant.

## 3. Results

Between July 2013 and January 2022, a total of 212 pregnancies were consulted in our center for fetal TE. Forty-eight pregnancies were excluded as further testing was refused. The mean maternal age was 29.8 (range 20.1–46.0) years, and the median gestational age of the fetus was 25.0 (range 12.7–33.3) weeks. The majority of patients were diagnosed in the second trimester (128/164 (78.0%)), 32 (19.5%) pregnancies were diagnosed in the last trimester, and the fewest number of patients, 2.4% (4/164), were diagnosed in the first trimester. Of these fetuses, 100 (61.0%) were male and 64 (39.0%) were female. This ratio was similar to that previously reported in the literature [4]. After diagnosis, 103 (62.8%) women chose to continue the pregnancy, while 55 (33.5%) chose to terminate of pregnancy (TOP), and six (3.7%) patients were lost to follow-up. The flowchart of genetic analysis progression is shown in Figure 1.

Among the 164 patients who met the inclusion criteria, CMA detected 17 (10.4%) clinically significant variants, including 16 (9.8%) fetuses with pathogenic copy number variant (pCNV) and 1 (0.6%) with likely pathogenic copy number variant (lpCNV), within which four fetuses were trisomy 18 (3/164) and mosaic trisomy 21 (1/164). Among 17 cases with clinically significant variants, there were 9 cases (52.9%) with CNVs < 10Mb, but another 8 cases (47.1%) were detected with CNVs > 10Mb. The most common CNV was the 22q11.2 microdeletion syndrome (*n* = 4). Non-isolated TE was combined with the most common abnormalities in the cardiovascular system (*n* = 4) and neurologic system (*n* = 4), and the most common was ventricular septal defect (*n* = 3). Table 1 shows the clinical and chromosomal characteristics of these 17 clinically significant variants.

Table 2 shows that the detection rate of CNVs in singleton pregnancies is significantly higher in non-isolated TE than in isolated TE (10/37, 27.0% vs. 6/111, 5.4%, *p* < 0.05). In twin pregnancies, 6.3% (1/16) were pCNV, which was not statistically different from singleton pregnancies (1/16, 6.3% vs. 16/148, 10.8%, *p* = 0.891). In terms of pregnancy outcomes, the rate of TOP was significantly higher in the non-isolated TE group than in the isolated TE group (26/37, 70.3% vs. 22/111, 19.8%, *p* < 0.05) and higher in the unilateral group than in the bilateral group (28/69, 40.6% vs. 20/79, 25.3%, *p* < 0.05), both of which were statistically significant.

We compared unilateral and bilateral foot for isolated and non-isolated TE in singleton pregnancies intergroup and intragroup (Figure 2). We found statistically significant differences in the detection rates of bilateral TE between isolated and non-isolated TE (2/63, 3.2% vs. 6/16, 37.5%, *p* < 0.05). CMA detected 25 cases of VOUS, but for financial reasons, some families rejected the suggestion to perform parental CMA verification because the price was close to $1000. Demographic and chromosomal data of VOUS are shown in Appendix A.

## 4. Discussion

In this study, we performed CMA on fetuses with an ultrasound diagnosis of TE and performed a follow-up evaluation to illuminate the genetic and clinical value of CNVs in fetal TE. We found that fetal TE with associated structural malformation correlates with a higher probability of clinically significant variants. The overall detection rate of clinically significant variants is similar to previous literature [14] (18/166, 10.8% vs. 17/164, 10.4%). In contrast, the proportion of CNVs that are undetectable by karyotyping (<10 Mb) among all clinically significant variants is nearly twice as high in our study (9/17, 52.9% vs. 5/18, 27.8%). Moreover, our study includes twin pregnancies and performs a comprehensive comparison of isolated and non-isolated TE, as well as unilateral and bilateral TE.

Talipes equinovarus is categorized into isolated and non-isolated TE. The isolated type is regarded as an isolated anomaly of the lower limbs that may be associated with polygenic inheritance; its prognosis is considered benign [15]. Non-isolated TE affects approximately 25% of fetuses and has been associated with deletion syndromes, aneuploidies, sex chromosomal abnormalities, neuromuscular diseases, microdeletions, and duplications [16]. In our cohort, there was a significant difference in the detection rate of isolated TE and non-isolated TE (6/111, 5.4% vs. 10/37, 27.0%, *p* < 0.05), a finding which is similar to previous studies [10].

We made comparisons of unilateral and bilateral TE in singleton pregnancies and found no significant differences (8/69, 11.6% vs. 8/79,10.1%, *p* = 0.774). Surprisingly, when comparing unilateral with bilateral foot for isolated and non-isolated TE in singleton pregnancies, we found a statistically significant difference in the detection rate of bilateral TE between isolated TE and non-isolated TE (2/63, 3.2% vs. 6/16, 37.5%, *p* < 0.05). This result suggested that when fetal TE is detected, in addition to excluding other structural abnormalities, it is crucial to preclude whether the contralateral foot also shows TE specifically. The reason is that chromosomal abnormalities are prevalent in bilateral TE. If other anomalies or bilateral TE are combined, we recommend further genetic testing.

Our data found a high incidence of 22q11.2 microdeletion syndrome (DiGeorge syndrome, DGS) in TE, which accounted for 23.5% (4/17) of all CNVs. Microdeletion of 22q11.2 is the most common microdeletion syndrome [17], and *TBX1* correlates with the most prominent phenotypes characteristic of this syndrome. Patients with 22q11.2 microdeletion syndrome display a broad array of phenotypes, and the most common findings include cardiac anomalies, hypocalcemia, and hypoplastic thymus. Case 10 in our non-isolated TE group was combined with ventricular septal defect, one of the common phenotypes above. Although skeletal anomalies are not a defining feature of DGS, studies reported that 1.1–13.3% of fetuses with this syndrome might have TE phenotype [17]. Interestingly, three of these four cases of fetuses with DGS were not combined with other structural abnormalities. To our knowledge, in the previous literature, there was only one case of a fetus with DGS with isolated TE detected by CMA, in Amihood et al. [10]. Unfortunately, these three cases all chose to termination of pregnancy and refused autopsy, so we cannot be sure if other common phenotypes of this syndrome were combined. No relevant literature has mentioned whether isolated TE is associated with the deletion of this fragment. We believe it is necessary to perform further studies to illustrate their correlation.

We included LOH in our study. LOH, also known as absence of heterozygosity, refers to long contiguous stretches of homozygosity in a chromosome. The pathogenesis of LOH includes homozygous mutation of recessive diseases and increased susceptibility to complex diseases [18,19], imprinting effects caused by uniparental disomy (UPD) [20], hidden mosaicism or confined placental mosaicisms [21], and potential association with tumorigenesis [22]. When LOH in the imprinting regions is confirmed to have been inherited from only one parent, it can cause imprinting disorders, such as Prader–Willi syndrome (maternal UPD15) and Angelman syndrome (paternal UPD15). Moreover, when LOH occurs on non-imprinted chromosomes, it may expose the causative gene of recessive genetic disorders. Liu et al. [23] reported that approximately 55% of LOH carriers had ultrasound abnormalities, and multiple malformations were the most common findings. In contrast, TE is generally dominantly inherited [24], but four of the five cases of LOH we identified occurred in isolated TE. Is this a coincidence, or is there a correlation between the two? More research is needed to answer this question.

In our data, case 3 was identified with a microduplication of 1.42 Mb in the chromosome 17p12 region, which was suggested to be pathogenic according to OMIM and DECIPHER databases. The clinical condition was Charcot–Marie–Tooth type 1A (CMT1A), associated with the PMP22 gene, and its phenotype is mainly characterized by multiple foot abnormalities and sensory abnormalities [25]. Notably, this case was a fetus with isolated TE, so it is reasonable to suspect that there is a more significant correlation with the genetic mechanism of TE, but this link remains to be explored. Case 14 was detected 25.38Mb duplication in chromosome 5p15.33p13.2, which contained 25 known OMIM disease genes. According to the DECIPHER and ClinGen databases, several studies have reported that patients carried segments of the presently detected segments and had a clinical phenotype that included TE [26].

It seems that several CNVs of non-isolated TE are occasional findings unrelated to foot malformations, such as Wolf–Hirschhorn syndrome, 21q22.13q22.3 microdeletion, et al. However, the clinical manifestations of these findings include hypotonia and sensory neurological dysfunction. Interestingly, among other anomalies combined with non-isolated TE, in addition to the cardiovascular system, neurological anomalies are the most common, such as porencephaly, holoprosencephaly, etc. Therefore, does this also support that some of the TE components are neuromuscular in origin? Martin et al. [27] revealed that defects in neuronal development caused by the overexpression of *Limk1* might lead to muscle atrophy and talipes equinovarus. Nevertheless, despite TE frequently occurring in neuromuscular abnormalities, no consistent neuromuscular abnormality is found in patients with isolated TE by electrophysiological examination or muscle biopsy [28,29,30].

In most cases, pregnancy outcome largely depended on whether prenatal ultrasound combined severe malformations and the chromosomal results of prenatal diagnosis. Parents always chose to terminate the pregnancy for pCNV with a poor prognosis, such as 22q11.2 microdeletion syndrome. For fetuses with VOUS, parents always chose TOP when subsequent ultrasound findings worsened. A negative result by CMA may help to increase parents’ confidence to continue the pregnancy. The pregnancy termination and live birth rates in cases with clinically significant variants detected by gene testing were found to be statistically significantly different from those with negative results (14/17, 82.4% vs. 41/147, 27.9%, *p* < 0.001). This suggests that the genetic test findings in fetuses of TE can affect parental decisions. In cases with clinically significant variants in genetic diagnosis, a small proportion of patients chose to continue the pregnancy after genetic counseling. In contrast, the majority chose termination of pregnancy due to the possible postnatal phenotype and poor prognosis. Therefore, it is necessary to perform genetic testing in patients with a prenatal ultrasound diagnosis of fetal TE, which may provide physicians and parents with more information about possible phenotypes after birth and to help them make pregnancy decisions and post-birth management.

There are some limitations to this study. First, this study is a retrospective study and the results lack of some crucial parameters, such as parental examination, family history, and results of genetic population screening tests such as spinal muscular atrophy. Second, fewer patients underwent karyotyping, and balanced translocations of chromosomes may have been missed. Third, although our study is the first prenatal study of singleton and twin pregnancies with TE using CMA in prenatal studies, the relatively limited number of twin pregnancies did not allow for definitive conclusions. We would like to collect more cases of twin pregnancies to further elucidate the potential differences between them. The final limitation is that TE may be related to a single gene or methylation. At the same time, we did not perform further tests such as whole exome sequencing (WES) and multiplex ligation-dependent probe amplification (MLPA) in our study.

## 5. Conclusions

This study is the most comprehensive prenatal study using CMA to perform a detailed molecular analysis of cases of fetal TE diagnosed in a Chinese population. Ultrasound detection of fetal TE may be a diagnostic clue for some fetal syndromes, and ultrasound abnormalities were associated with increased risk of CMA findings in both singleton and twin pregnancies and both isolated and non-isolated TE. Therefore, when fetal TE is diagnosed by prenatal ultrasound, attention should be paid to whether it is combined with other structural abnormalities and to further genetic testing that could exclude these genetic disorders, which may help with diagnosis and counseling of prenatal TE.

## Figures and Tables

**Figure 1 genes-13-01573-f001:**
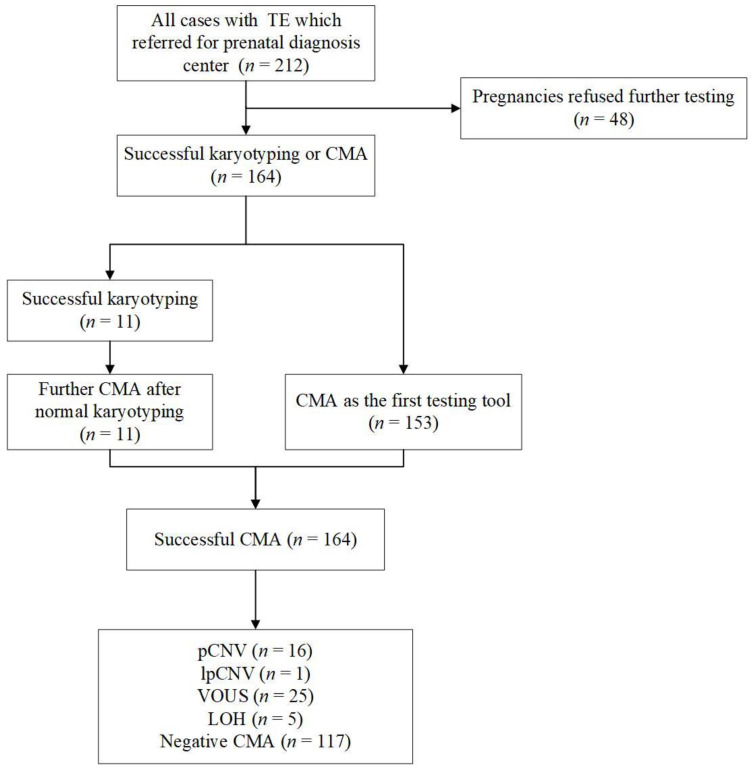
Flowchart of genetic analysis progression in cohort of fetuses with TE.

**Figure 2 genes-13-01573-f002:**
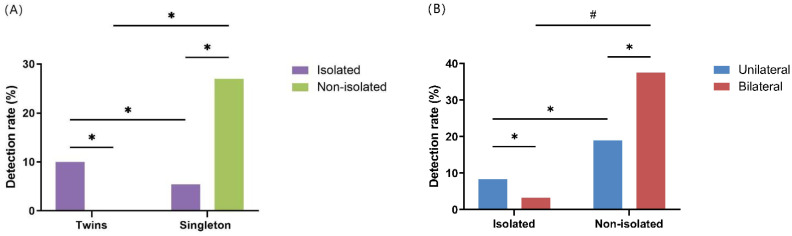
Comparison of CMA detection rates of fetuses with talipes equinovarus. (**A**) Comparison of CMA detection rate of isolated and non-isolated TE in fetuses with singleton and twin pregnancies; (**B**) Comparison of CMA detection rate of unilateral and bilateral TE in fetuses with isolated and non-isolated TE. * *p* > 0.05; # *p* < 0.05.

**Table 1 genes-13-01573-t001:** Clinical features in fetuses with talipes equinovarus.

Case	Maternal Age (years)	GA (weeks)	Ultrasound Findings	CMA Results	Type of CNV	Size (Mb)	Outcome
**1 ***	29.6	30.6	Isolated TE	arrXp22.31(6,449,752–8,143,319) × 1	Deletion	1.69	Live birth
**2**	39.9	18.9	Isolated TE	arr22q11.21(18,636,749−21,800,471) × 1	Deletion	3.16	TOP
**3**	20.1	28	Isolated TE	arr17p12(14,087,918−15,503,234) × 3	Duplication	1.42	TOP
**4**	30.3	26	Isolated TE	arr22q11.21(18,916,842−21,800,471) × 1	Deletion	2.88	TOP
**5**	34.9	28.6	Isolated TE	arr18q21.32q23(57,600,965−78,014,123) × 1	Deletion	20.41	TOP
**6**	28.2	28.4	Isolated TE	arr22q11.21(18,916,842−21,465,662) × 1	Deletion	2.55	TOP
**7**	26.4	24	Isolated TE	arr(21) × 2~3	Duplication	33.09	TOP
**8**	27.9	26.4	TE; VSD	arr4p16.3p15.33(68,345−14,195,870) × 1	Deletion	14.13	TOP
**9**	39.3	20.7	TE; CPCs	arr(18) × 3	Duplication	77.88	TOP
**10**	31.1	19.6	TE; VSD	arr22q11.21(18,916,842−21,465,662) × 1	Deletion	2.55	TOP
**11**	26.0	19	TE; CLP; HPE; SGA	arr6p25.3p24.3(156,975−9,116,357) × 3 arr21q22.13q22.3(38,242,327−48,093,361) × 1	Duplication Deletion	8.96 9.85	TOP
**12 #**	29.2	33	TE; oligohydramnios	arr22q11.21(20,717,654−21,465,659) × 1	Deletion	0.75	Live birth
**13**	28.9	24.6	TE; cholecystomegaly	arr16p13.11(14,896,385−16,328,840) × 3	Duplication	1.43	Live birth
**14**	28.6	25.7	TE; iCTR	arr5p15.33p13.2(2,103,059−37,483,088) × 3	Duplication	25.38	TOP
**15**	35.9	25	TE; VSD	arr(18) × 3	Duplication	77.88	TOP
**16**	32.9	23.4	TE; FGR; porencephaly	arr13q22.1q33.1(74,307,209−102,461,029) × 1	Deletion	28.15	TOP
**17**	31.5	12.7	TE; CH; omphalocele	arr(18) × 3	Duplication	77.88	TOP

*: one of twin fetuses; #: likely pathogenic CNV; TE: talipes equinovarus; VSD: ventricular septal defect; CPCs: choroid plexus cysts; CLP: cleft lip and palate; HPE: holoprosencephaly; SGA: small for gestational age; iCTR: increased cardiothoracic ratio; FGR: fetal growth restriction; CH: cystic hygroma; TOP: termination of pregnancy.

**Table 2 genes-13-01573-t002:** Stratified analysis of CNVs detection and pregnancy outcome in TE.

Groups	CSVs	VOUS	Live Birth	TOP
**Singleton vs. Twins**				
Singleton (*n* = 148)	16 (10.8%)	24 (16.2%)	95 (64.2%)	48 (32.4%)
Twins (*n* = 16)	1 (6.3%)	1 (6.3%)	8 (50.0%)	7 (43.8%)
*p*-value	0.891	0.492	0.265	0.362
**Isolated vs. Non-isolated ***				
Isolated (*n* = 111)	6 (5.4%)	19 (17.1%)	85 (76.6%)	22 (19.8%)
Non-isolated (*n* = 37)	10 (27.0%)	5 (13.5%)	10 (27.0%)	26 (70.3%)
*p*-value	0.000	0.607	0.000	0.000
**Unilateral vs. Bilateral ***				
Unilateral (*n* = 69)	8 (11.6%)	14 (20.3%)	38 (55.1%)	28 (40.6%)
Bilateral (*n* = 79)	8 (10.1%)	10 (12.7%)	57 (72.2%)	20 (25.3%)
*p*-value	0.774	0.209	0.031	0.048
**Left foot vs. Right foot ***				
Left foot (*n* = 32)	2 (6.3%)	5 (15.6%)	20 (62,5%)	12 (37.5%)
Right foot (*n* = 37)	6 (16.2%)	9 (24.3%)	18 (48.6%)	16 (43.2%)
*p*-value	0.362	0.370	0.249	0.752

*: Comparison in singleton pregnancies; TE: talipes equinovarus; CSVs: clinically significant variants; VOUS: variants of unknown significance; TOP: termination of pregnancy.

## Data Availability

The data that support the findings of this study are not publicly available as the information contained could compromise the privacy of research participants. Further inquiries can be directed to the corresponding author.

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
