# Peer review of "Prenatal Diagnosis of Talipes Equinovarus by Ultrasound and Chromosomal Microarray Analysis: A Chinese Single-Center Retrospective Study"

_genes, 2022, doi:10.3390/genes13091573_

Round 1
Reviewer 1 Report
Talipes equinovarus (TE), also known as clubfoot is a well-recognized congenital foot deformity. The manuscript presents the results of chromosomal microarray analysis (CMA) in a relatively large cohort of pregnancies with isolated and non-isolated talipes equinovarus.
Of the 164 CMA tests, a total of 17 (10.4%) clinically significant variants were detected. Of the 229 cases with isolated deformity, nine (3.9%) clinically significant CMA results were detected. the diagnostic rate of clinically significant variants was significantly higher in the non-isolated TE group than in the isolated TE group (27.0% vs. 5.4%, P < 0.001).
The authors demonstrated that CMA is useful in prenatal diagnosis of TE . Moreover, prenatal ultrasound detection of clubfoot associated with additional sonographic defects further increases the risk for abnormal CMA findings.
Points:
Introduction
1. Introduction provides basic information on TE epidemiology. The incidence rate must be corrected ( 1-3 cases per 1000, not 10000).
2. The ultrasound diagnosis is not restricted to second and third trimester ( see systematic review by Laura Ruzzini, Diagnostics 2021).
3. There are several important publications on CMA utility in TE -both prenatal and postnatal cases ( Alvarado DM., 2013; Amihood S., 2020). They should be included in the introduction ( and/or discussion).
Methods
4. Could you please specify why informed consent is not required in this case?
Results
5. When presenting CMA results you need to provide reference. Which human genome assembly was used hg19 or hg38?
6. According to ISCN trisomy 18 when detected by CMA should be reported as: arr(18)x3
Author Response
Point 1: Introduction provides basic information on TE epidemiology. The incidence rate must be corrected (1-3 cases per 1000, not 10000).
Response 1: Thank you for pointing out this error. We have corrected it in this revision. (Page 1, lines 38)
Point 2: The ultrasound diagnosis is not restricted to second and third trimester (see systematic review by Laura Ruzzini, Diagnostics 2021).
Response 2: We sincerely appreciate the valuable comments. We have checked the relevant literature carefully. As ultrasound equipment and technology have improved, some fetuses with clubfoot can be diagnosed in early pregnancy, so we have made revisions based on your comment and have cited the appropriate literature. (Page 1, lines 41)
Point 3: There are several important publications on CMA utility in TE -both prenatal and postnatal cases (Alvarado DM., 2013; Amihood S., 2020). They should be included in the introduction (and/or discussion).
Response 3: Thank you for your constructive suggestions. We have read these two representative articles carefully and think it is necessary to put them in the introduction section to introduce the use of CMA in prenatal and postnatal TE. We have added it to the introduction and discussion section of the article following your recommendation. (Page 2, lines 55-61)
Point 4: Could you please specify why informed consent is not required in this case?
Response 4: Thank you very much for your question. Our previous sentence might cause ambiguity. We intended to express that as a retrospective study, there is now no necessity to ask for patient consent again when reviewing medical records and using them for publication. This is because before performing genetic testing, each patient will sign a written informed consent form and agree to the data being used for subsequent scientific research and publication. We have revised this expression in the revised version. We have apologized for the misunderstanding we caused and thank you again for your constructive suggestion. (Page 2, lines 88-89)
Point 5: When presenting CMA results you need to provide reference. Which human genome assembly was used hg19 or hg38?
Response 5: Thank you for this insightful comment. The built reference genome was aligned on GRCh37/hg19. We have added it in the relevant section of our article. Thank you for your timely notification so that we can avoid any misunderstanding that might have resulted. (Page 2-3, lines 98-99)
Point 6: According to ISCN trisomy 18 when detected by CMA should be reported as: arr(18)x3
Response 6: Thank you very much for your careful reading. We have corrected it in this revision following ISCN 2020. (Table 1, case 7,9,15,17)

Reviewer 2 Report
Dear Editor,
Thank you for giving me the opportunity to review this interesting article on the value of chromosomal microarray analysis in fetuses affected by congenital talipes.
I found the article well written with consistency between the different components of the text. However, I would be a bit more specific in the title since it is not very clear that the study is focused on the additional value of CMA analysis in fetuses with talipes.
Apart from this aspect, I only have some minor comments I would like to address to the authors:
· Page 2, lines 58-60: I would move this sentence to the discussion paragraph or leave it to the conclusion were the authors already put it.
· Page 2, line 84: I think that “parental blood” cannot be an option taken singularly for genetic test. I believe the authors should rephrase the sentence by saying something like “Fetal genomic DNA was extracted from chorionic villi, amniocytes or cord blood. Informed consent was taken to obtain a parental blood sample in order to run a trio analysis”. Otherwise, it looks like parental blood may have been taken alone without a fetal sample.
· Methods paragraph: later in the discussion section the authors refer to the performance of loss of heterozygosity analysis, but there is no mention in the methods paragraphs. Please add a sentence that explains if it was performed on all cases or only upon specific indications.
· Page 7, line 203-207: I just have a comment on this aspect. The authors found that among 4 cases out of 5 with loss of heterozygosity the fetal talipes was isolated. As mentioned by authors, fetal talipes, when isolated and usually benign, may have a familiar tract even in the absence of genetic abnormalities. May the presence with loss of heterozygosity have a causative link with recurrent familiar form of isolated talipes in the opinion of the authors? Do they have family history in this case to explore the presence of familiar recurrence?
· Figure 2. The finding that more women carrying a fetus with unilateral TEV includes the presence of associated anomalies? By looking at Figure 2B it looks like the number of unilateral talipes in the non-isolated forms seems to be the highest. I believe that this aspect should be pointed out in the discussion in order to explain why more women with unilateral talipes underwent termination compared to bilateral forms, which seems illogic.
· I think that the number of twin pregnancy is too low to draw definitive conclusions on TEV in singletons versus twins pregnancies and this aspect should be highlighted by authors as a limit of the study.
Author Response
Point 1: “However, I would be a bit more specific in the title since it is not very clear that the study is focused on the additional value of CMA analysis in fetuses with talipes.”
Response 1: We sincerely appreciate the valuable comments. The original title was indeed ambiguous and we have changed the title to “Prenatal diagnosis of talipes equinovarus by ultrasound and chromosomal microarray analysis: a Chinese single-center retrospective study” based on your suggestion. (Page 1, lines 2-3)
Point 2: “Page 2, lines 58-60: I would move this sentence to the discussion paragraph or leave it to the conclusion were the authors already put it.”
Response 2: Thank you for this insightful comment. We have already emphasized this innovation in the conclusion section, so we decided to delete this sentence here to avoid repetition. (Page 2, lines 66-68)
Point 3: “Page 2, line 84: I think that “parental blood” cannot be an option taken singularly for genetic test. I believe the authors should rephrase the sentence by saying something like “Fetal genomic DNA was extracted from chorionic villi, amniocytes or cord blood. Informed consent was taken to obtain a parental blood sample in order to run a trio analysis”. Otherwise, it looks like parental blood may have been taken alone without a fetal sample.”
Response 3: Thank you for your constructive suggestions. Our previous sentence might cause ambiguity. We have made changes based on your suggestions. Please allow us to express my appreciation again for your careful reading. (Page 2, lines 93, 95-96)
Point 4: “Methods paragraph: later in the discussion section the authors refer to the performance of loss of heterozygosity analysis, but there is no mention in the methods paragraphs. Please add a sentence that explains if it was performed on all cases or only upon specific indications.”
Response 4: Thank you very much for your comment. The CytoScan 750K or CytoScan HD microarrays will routinely detect LOH and UPD, and we have added an explanation of this in the methods section, thanks again for your advice. (Page 3, lines 99-102)
Point 5: “Page 7, line 203-207: I just have a comment on this aspect. The authors found that among 4 cases out of 5 with loss of heterozygosity the fetal talipes was isolated. As mentioned by authors, fetal talipes, when isolated and usually benign, may have a familiar tract even in the absence of genetic abnormalities. May the presence with loss of heterozygosity have a causative link with recurrent familiar form of isolated talipes in the opinion of the authors? Do they have family history in this case to explore the presence of familiar recurrence?”
Response 5: We feel great thanks for your professional review work on our article. This is a very constructive question, and we have previously taken this issue into consideration. But unfortunately, none of the parents of these 5 fetuses had the phenotype of TE after our follow-up. Therefore, we cannot draw a conclusion on it, but we suppose that there may be a relationship between them, and we propose it here and hope that more studies will confirm this speculation.
Point 6: “Figure 2. The finding that more women carrying a fetus with unilateral TEV includes the presence of associated anomalies? By looking at Figure 2B it looks like the number of unilateral talipes in the non-isolated forms seems to be the highest. I believe that this aspect should be pointed out in the discussion in order to explain why more women with unilateral talipes underwent termination compared to bilateral forms, which seems illogic.”
Response 6: Thank you very much for such a constructive question. When we saw this question, we thought it was a very interesting phenomenon, so we immediately read the entire article again and reviewed all the statistical data. Finally, we found that the correspondence of the Figure 2B icons was reversed, that is, "blue-unilateral; red-bilateral" was replaced by "red-unilateral; blue-bilateral". We again checked the detection rates for each subgroup and the statistics were as follows: isolated unilateral TE (4/48, 8.3%), isolated bilateral TE (2/63,3.2%), and non-isolated unilateral TE (4/21, 19.0%), and non-isolated bilateral TE (6/16, 37.5%). The statistical data and results analysis were both correct, but the icons were reversed. We have corrected it in the latest revision. We do apologize for our carelessness and thank you for your timely notification so that we can avoid any misunderstanding that might have resulted. (Figure 2B)
Point 7: “I think that the number of twin pregnancy is too low to draw definitive conclusions on TEV in singletons versus twins pregnancies and this aspect should be highlighted by authors as a limit of the study.”
Response 7: Thank you for your constructive suggestions. We have added this item to the limitations section based on your advice. (Page 9, lines 262-265)
